# Convenient Real-Time Monitoring of the Contamination of Surface Ion Trap

**DOI:** 10.3390/nano10010109

**Published:** 2020-01-06

**Authors:** Xinfang Zhang, Yizhu Hou, Ting Chen, Wei Wu, Pingxing Chen

**Affiliations:** 1Department of Physics, College of Liberal Arts and Sciences, National University of Defense Technology, Changsha 410073, China; xfz-6446@nudt.edu.cn (X.Z.); houyizhu@nudt.edu.cn (Y.H.); chenting8522@163.com (T.C.); 2Interdisciplinary Center for Quantum Information, National University of Defense Technology, Changsha 410073, China

**Keywords:** adatoms contamination, thin film, surface ion trap, anomalous heating of ions, quantum information processing

## Abstract

Recent studies indicated that contamination by adatoms on the surface ion trap can generate contact potential, leading to fluctuations in patch potential. By investigating contamination induced by surface adatoms during a loading process, a direct physical image of the contamination process and the relationship between the capacitance change and the contamination from surface adatoms is examined theoretically and experimentally. From the relationship, the contamination by surface adatoms and the effect of in situ treatment process can be monitored by the capacitance between electrodes in real time. This study is foundational to further research on anomalous heating with practical applications in quantum information processing from surface ion traps.

## 1. Introduction

Anomalous heating of ions, frequently observed experimentally, is a major obstacle in quantum information processing (QIP) from the surface ion trap [1,2,3,4]; however, thus far, this effect is not completely understood. Studies have demonstrated that fluctuating patch potentials on the electrode comprise a major source of unexpected large heating rates [2,3,5,6]. Patch potential can be caused by light induced charge on the substrate and surface adsorption on electrodes [5]. In general, light induced charge discharges after a few days [5,7]. By contrast, surface adsorption of atoms and molecules, particularly atoms induced in the loading process, is maintained over time [5,8,9].

Previous experiments have demonstrated that trap surface adatoms from the exposure of the atomic beam generate strong electric stray fields [9,10], in particular on the loading region of ions on the surface ion trap [5]. The electric stray field continuously changes by V/m over three months [11], causing high heating rates. From recent experiments, in situ trap treatments can be used to remove surface adsorption [6,12]. The treatments include ion milling, pulsed-laser cleaning, and plasma treatment, which reduce the heating rate by approximately 100 fold [13], 4 fold [7], and 3 fold [14], respectively. Although, the treatment processes are useful for surface ion traps, there are inevitable limitations [14]. Ion milling causes redeposition and engineering complexities in experimental vacuum systems [7]. Pulse-laser cleaning causes visible damage and light induced charges on the substrate. Plasma cleaning is used to remove hydrocarbons from the surface [14]. To prevent damage to the surface ion trap, the number of cleaning processes is regulated. Thus, identifying the cleaning time and effect on the trap surface is vital to protect the surface ion trap.

In this study, we investigate the contamination by surface adatoms induced in the ion loading process. The capacitance between electrodes is examined by testing ten trap samples. The capacitance measurement, which can be directly and conveniently measured, can describe the trap contamination from surface adatoms. Precisely, with monolayer adatoms covering the trap surface, the capacitance between electrodes increases by approximately tenths of a picofarad. The capacitance between electrodes, however, does not change before and after the deposition of monolayer atoms. Furthermore, we describe the contamination process from the simulation. Further experiments using atomic force microscopy (AFM) and scanning electron microscopy (SEM) are conducted, and variations in the capacitance between electrodes are examined. We propose a simple method of capacitance measurement to estimate contamination by surface adatoms.

## 2. Experiment and Simulation of Trap Surface Adatoms

Our ion loading scheme is shown in Figure 1a and is widely used in surface ion traps [4,6,15,16,17], in particular traps made of non-silicon substrate materials. Atoms leave the atomic oven with a solid angle and form an atomic beam, which sweeps across the surface ion trap, resulting in adsorption of some atoms on the surface, hence contaminating the trap. Numerous studies indicated that trapped ions suffer from the effect of surface adatoms [5,7,11,18], which will influence the ion trapping stability. Figure 1b shows a circuit diagram of a surface ion trap. Here, capacitance between the two adjacent electrodes is monitored when the atomic oven is turned on.

### 2.1. Experimental Setup and Results

Our surface ion trap was based on gold-on-silica fabrication technology and a 10μm-wide gap between the 2.5μm thick gold electrodes, as shown in Figure 1a. The experimental system comprised a vacuum chamber, a calcium (Ca) oven, and a surface ion trap. The radio frequency (RF) electrode and ground electrode were connected to a two-pin electric feedthrough to monitor capacitance variations using an inductance-capacitance-resistance (ICR) tester, as shown Figure 1b. The horizontal and vertical distances from the calcium oven to the trap center were approximate 4 cm and 1.5 cm, respectively. The atomic beam generated by the atomic oven swept across the trap surface. The calcium vapor pressure and temperature of the atomic oven were adjustable. The total number of surface adatoms increased if the atomic oven was turned on. The temperature of the Ca atomic oven was measured using an attached thermocouple. The calcium vapor pressure at the trap position was measured using a residual gas analyzer (RGA).

We examined the influence of turning on the atomic oven on the capacitance between the electrodes. In the experiment, the temperature of the atomic oven and the corresponding Ca vapor pressure at the trap position were set to 700 K and approximately $5\times 10^{-4}$ Pa, respectively, when the atomic oven was turned on. We turned off the atomic oven after 5 min. The capacitance between RF and ground electrodes was recorded during reduction of the pressure of chamber to $10^{-7}$ Pa. This operation was repeated several times (turning on and off, waiting, measuring), to record electrode capacitances at different times of turning on the atomic oven. We used ten of the same trap samples to repeat the same operations and obtained the average of the capacitance. Finally, we determined the relationship between electrode capacitance and time of turning on the atomic oven, as shown in Figure 2, with the solid line being a visual guide. Each sample’s capacitance was about 2.5 pF before turning on the atomic oven. Even dozens of minutes after the initial exposure, the capacitance between the electrodes remained basically the same. Surprisingly, the capacitance rose sharply after 70 min of turning on the oven and stabilized after 90 min. The maximum capacitance difference was approximately 0.2 pF. The capacitance changes over time are marked by the grey area in Figure 2. The results revealed that the capacitance change between electrodes was closely related to the contamination from surface adatoms. Furthermore, we inferred that the capacitance changed if the stage of the monolayer film resulted in the formation of adatoms.

### 2.2. Theoretical Model and Numerical Simulation

To understand the surface adatoms’ process, the theoretical model and numerical simulation were established. In a high vacuum, the properties of atomic gas obey the state equation of an ideal gas that follows the Boltzmann distribution [19]. At a pressure *P*, we considered atoms of mass *m* leaving the oven at a temperature *T* with a solid angle. Atoms leaving the oven spread over the trap surface with distance *r* at an angle θ and normal angle β. The parameters θ, β, and *r* were obtained by measurements from the relative spatial position of the surface ion trap and the atomic oven. The distribution of atoms on the surface area *S* [20] is given by:(1)$$D=\int_{S}\frac{\Phi cos\theta cos\beta }{\pi r^{2}}dS,$$
where Φ is the leaving rate from the oven per unit area and unit time, i.e., the outgoing flux defined by:(2)$$\Phi =\alpha {\left(\frac{N_{A}^{2}}{2\pi mRT}\right)}^{\frac{1}{2}}P.$$
where α is the evaporation coefficient [21], $N_{A}$ is Avogadro’s number, and *R* is the universal gas constant.

Surface adatoms were simulated numerically by the finite element method and were characterized by AFM and SEM. The thick adatom films were highly visible and reduced the influence of the surface roughness of the electrode or substrate. This increased the accuracy of measuring film thicknesses. To examine the correctness of the theoretical model to characterize adatom films, we adopted high temperatures up to 900 K to complete the adatoms’ experiments rapidly with corresponding Ca vapor pressure at the trap position set to $3\times 10^{-3}$ Pa. These parameters were used in simulations with different thicknesses of films on the surface. The simulation results in Figure 3 show that the width of the different thicknesses of films was invariably 3.7μm, covering one side of the gap. There were no adatoms on the other side of the electrodes’ gap. The 3.7μm-wide adatom film connected to only one side of the electrodes. Furthermore, the film thickness was linearly dependent on the exposure time of the atomic beam, as shown in Figure 3.

To compare with the simulation, we performed the surface adatoms’ experiment for 13.5 h, with parameters consistent with those of the simulation. The sample covering the Ca film was tested using AFM with Ca film thickness of approximately 50 nm, as shown in Figure 4. Because of the limited measurement range of the AFM, the entire electrode gap could not be scanned. The SEM image of the trap surface showed that the 10μm-wide gap of the electrodes was covered by 3.7μm wide Ca film that connected to one side of the electrodes, as shown in Figure 5. These results agreed with the simulation, demonstrating the reliability of our calculation. Changes in capacitance between the electrodes was simulated using COMSOL software (version 5.1, Stockholm, Sweden) for electromagnetic analysis. Based on the geometry of the RF electrode and the ground electrode on the silica substrate, the capacitance was simulated with 1 V applied to the RF electrode and the ground electrode at 0 V. The simulated capacitance difference with and without the 3.7μm wide adatom film was 0.2 pF, consistent with our experimental results.

According to the parameters of our experiment, we simulated the surface adatoms with the atomic oven turned on. The Ca monolayer by simulation was formed in about 65 min. This approximated the starting time for capacitance changes. Furthermore, there was a temperature response process associated with the atomic oven when turned on and off in the experiment. This led to shorter forming time of the monolayer film by simulation compared to that of the experiment.

We can see that the capacitance started to increase sharply after the atomic oven was turned on for 70 min, as shown in Figure 2. The increase in capacitance indicated that the monolayer film was forming owing to surface adatoms on the surface ion trap. Therefore, the capacitance change was owed to structural reorganization of the trap electrode resulting from the adsorbed film on the electrode gap attached to one side of the electrode. The monolayer fully formed 90 min after turning on the atomic oven. Because there was only the variation of nanoscale thickness, negligible for an electrode several micrometers thick, the capacitance between the electrodes did not change after 90 min after turning on the atomic oven.

To estimate the forming time of the monolayer adatom film for the ion QIP experiment, we simulated the surface adatoms induced in the ion loading process under $10^{-8}$ Pa vacuum pressure and 500 K temperature. By continuously turning on the atomic oven, we demonstrated that the time for forming a monolayer was approximate three months. If the atomic oven was turned on for three hours each day in the experiment, the surface ion trap could be used for two years before contamination from monolayer adatoms, coinciding with the time of unstably trapping ions in [8].

The formation rate of adatom films depended on the temperature of the atomic oven. We considered the cases of high, medium, and low temperature. Using the high temperature, 900 K, of the atomic oven that gave the rapid formation of a thick film, the simulations were compared to our experiments to determine a reliable theoretical model. At the medium temperature, 700 K, the capacitance change trend was quickly measured. The measurement was consistent with the corresponding simulated results from our theoretical model. Generally, during the ion loading process, the Ca atomic oven employed a low temperature, about 500 K, in the QIP experiment, and we determined the corresponding forming time of the adatoms monolayer on the surface ion trap.

## 3. Analysis and Discussion

We inferred from our study the physical image of the contamination process. In the ion loading process, the trap surface was exposed to an atomic beam. Atoms were adsorbed and randomly distributed on the surface. A monolayer film covering the electrode gap was connected to one side of the electrode and resulted in a reconstructed electrode and changes in capacitance between the electrodes. Although the adatom films became thicker during the ion loading process, the electrode capacitance did not change. Atomic ovens were foundational to the study of the surface adatoms processes. Furthermore, it was suitable for laser ablation loading ions in a surface ion trap [22].

The monitoring of the capacitance between electrodes could reflect the surface adatoms with the oven turned on in three stages: (I) a single atom is randomly distributed on the surface, and the capacitance remains at its initial value; (II) a monolayer film is formed gradually, and the capacitance between the electrodes starts to change; (III) the adatom film thickens, and the capacitance stays constant. This contamination measuring method was an intuitive and simple method to estimate surface adatoms in real-time. It was effective for the simple structures of the surface ion trap. Because the measurement of electric field noise from trapped ions is complex, surface adatoms and other factors accounted for electric field noise. Although electric-field noise can be measured by trapped ions [6,9,23,24] to describe surface adatoms indirectly, it is arduous to monitor in real-time to reflect adatom films. Our contamination measuring method for a surface ion trap was foundational to further studies on the anomalous heating of ions.

Because trapped ions are affected by a high heating rate in a surface ion trap, the trap surface requires cleaning if a monolayer is formed, although the trap can barely trap ions at that stage [11]. In the process of in situ treatments, the monitoring of capacitance between the electrodes can reflect the treatment effect preventing over cleaning, which may damage the surface ion trap and produce undesirable effects [14].

## 4. Conclusions

In this study, we obtained a relationship between the capacitance change and the contamination from surface adatoms through theory and experimentation. From this relationship, we measured the contamination by surface adatoms and the effect of the in situ treatment process by monitoring the capacitance between electrodes in real-time. This was remarkable because it permitted steady ion trapping.

## Figures and Tables

**Figure 1 nanomaterials-10-00109-f001:**
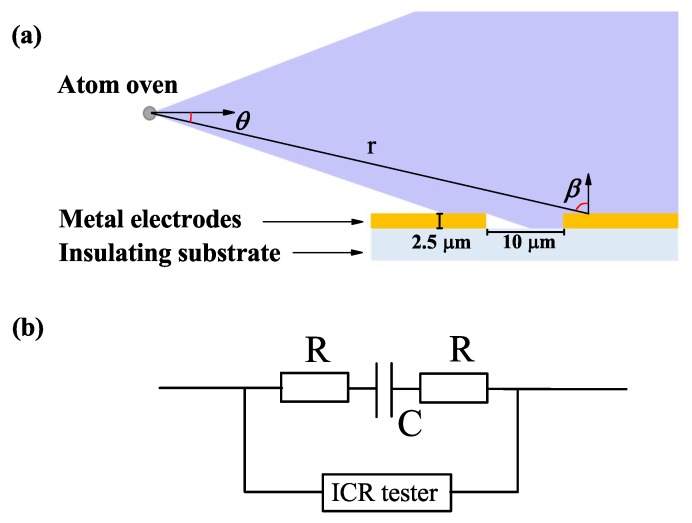
(Color online) (**a**) Cut through the atomic beam with solid angle sweeps across the surface ion trap. Atoms leaving the oven through an angle θ are spread over the trap surface with distance *r* and normal angle β. (**b**) Circuit diagram of the surface ion trap. The electrode gap is an equivalent capacitance, *C*, and the electrode is an equivalent resistance, *R*.

**Figure 2 nanomaterials-10-00109-f002:**
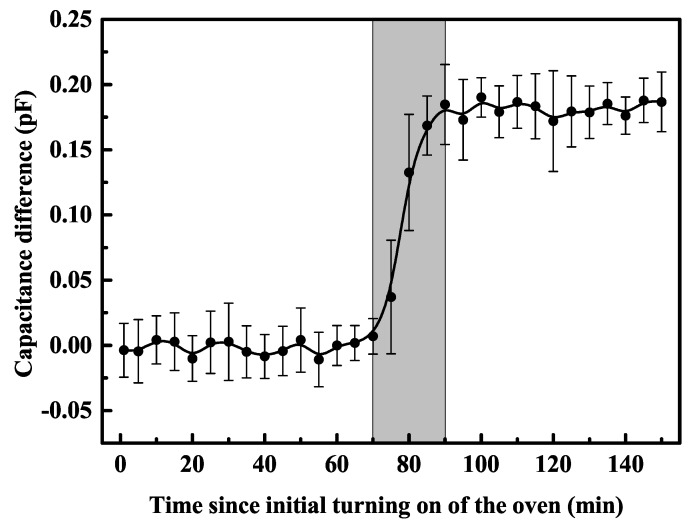
Variation of the capacitance from ten identical trap samples measured over time after turning on the atomic oven, under the Ca vapor pressure $5\times 10^{-4}$ Pa and temperature 700 K of the atomic oven. The values of the data points are the average of ten trap samples’ capacitances starting when the oven was turned on (t=0 min) and continuing after the oven was turned off at t=5 min. After the oven is turned off, the vacuum returns to $10^{-7}$ Pa. The maximal capacitance difference is approximately 0.2 pF.

**Figure 3 nanomaterials-10-00109-f003:**
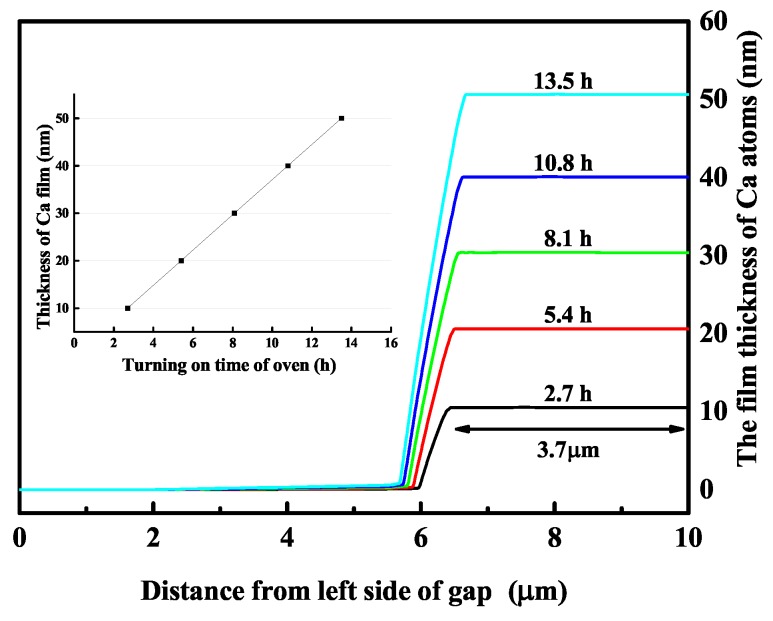
(Color online) The different adatoms’ thicknesses on the electrode gap are calculated at Ca vapor pressure $3\times 10^{-3}$ Pa and the temperature 900 K of the atomic oven. (10,20,30,40,50) nm thick films are generated by turning on the atomic oven for (2.7,5.4,8.1,10.8,13.5) h, and the widths of these adatoms films are approximately 3.7μm. The different thicknesses of the adsorbent layer are linearly dependent on the exposure time to the atomic beam in the illustration.

**Figure 4 nanomaterials-10-00109-f004:**
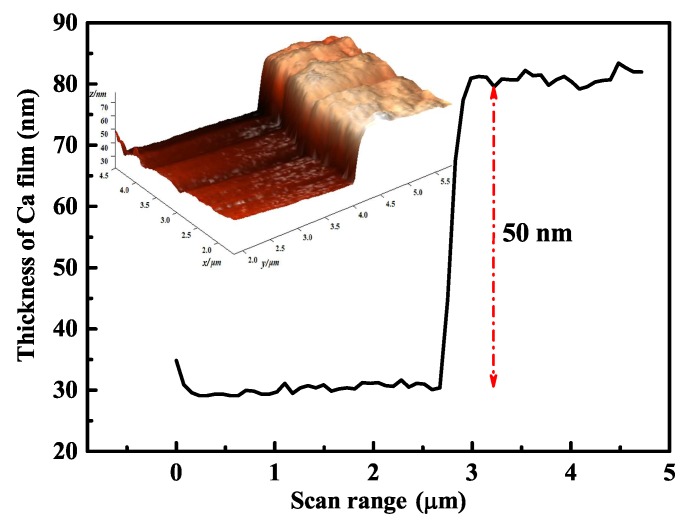
(Color online) The 50 nm thickness of the adsorbed atom film on the gap of electrodes tested by AFM, which comes from the AFM 3D image. There is a sharp increase at the 2.7μm position of the scan range, signifying the edge of the Ca atomic film.

**Figure 5 nanomaterials-10-00109-f005:**
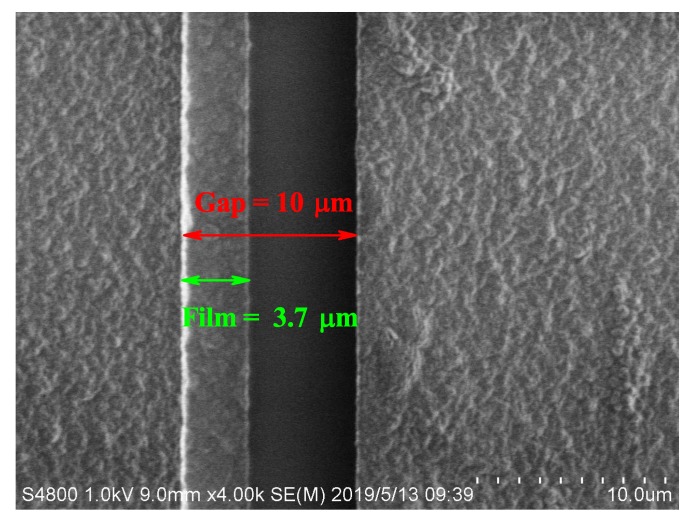
(Color online) SEM images of the surface ion trap with adsorbed atoms. The gap width of the two electrodes is approximately 10μm covering a Ca atomic film with a 3.7μm width.

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
