# Peer review of "Convenient Real-Time Monitoring of the Contamination of Surface Ion Trap"

_nanomaterials, 2020, doi:10.3390/nano10010109_

Round 1
Reviewer 1 Report
This paper presents results from an investigation of adatoms contamination on surface traps. I think continued study of this experimental challenge is an interesting and important topic for use of these devices for QC/QS. While certainly not the first time this has been noted or studied, I found the comparison of the simulation to the atomic force microscope and scanning tunneling microscope images informative and potentially useful for developing schemes for minimizing this effect in future experiments.
Some of the text should be corrected for clarity.
1) The comparison between the simulation and images is impressive. I would like to know more how the authors determined theta (the dispersion angle from the oven) and whether r (distance from the oven) was measured or a free parameter in the simulation. And in equation is alpha (the evaporation coefficient) determined calculable or fit?
2) I am confused by the experimental procedure in Sec 2.1. My understanding from the text is that in each trial the oven is turned OFF after 5 minutes and it takes subsequent 65 minutes for the capacitance to change (while pressure in the chamber is going down). I understand that the x-axis in Figure 1 starts with t=0 being the initial turning on of the oven, but I find the label misleading (if I have understood the experiment correctly). I suggest “Time since initial turning on of the oven” and correcting sentence 2 of the caption to “The values of the data points are the average of ten trap samples’ capacitances starting when the oven was turned on (t=0 min) and continuing after the oven was turned off at t= 5 mins. After the oven is turned off the vacuum returns to 10−7 Pa.”
With regard to this procedure I don’t understand line 73 “When turning on the atomic oven for dozens of minutes” I think it should be “Even dozens of minutes after the initial exposure, the capacitance between the electrodes remains basically the same.”
3) I find the x-label for Figure 3 also initially confusing. It seems like an appropriate label if one was varying the gap size. I think a better choice may be “ Distance from left side of gap (um)”
In summary, I think it was interesting to see the coupling of the simulation and the atomic force microscope and scanning tunneling microscope images. The simulation is clearly able to capture the time dynamics of the growth and the spatial extent. Measuring the capacitance seems an easy way to detect this growth although the waiting time of 70 minutes seems to make this less a real time solution but perhaps a useful diagnostic none the less. I think it is worth publishing but the English writing could use another read through by native speaker. I have listed some additional typos to correct below for ease of reading.
Additional typos.
line 2= add s to trap
line 23: several misspelled
line27: “to” out of place should be “up to roughly”
line36: should be “testing”
line 45: remove to load ions (redundant) should be “Our ion loading scheme is shown in “
line 53: In my opinion the greek mu should be used here (and elsewhere) for the unit in latex $\mu$
line 68: after “samples, “ the word We should be lower case.
line 69: the sentence “The initial capacitance between electrodes is about 2.5 pF is redundant to line 72 and should be removed.
line 75: “maximum” is misspelled.
line 79: monolayer film results in the formation of adamtoms
line 95: period at the end should be a comma “experiment faster, and the corresponding”
line 99: replace another with “the other”
line 100: “connect to only one side”
line102: replace “according” with “To compare to the “
Figure 3 caption different in last line is misspelled.
Figure 4 caption I am not sure what word “precess” is meaning. I suggest “There is a sharp increase at the 2.7um position of the scan range, signifying the edge of the adaptors film.”
line 146: missing NOT and capacitance. “, the electrode capacitance will not be changed anymore.”
Reviewer 2 Report
The authors present a simple technique for monitoring the contamination of a surface electrode ion trap. By measuring the capacitance between neighboring trap electrodes, they are able to detect surface contamination caused by adatoms from a Ca oven. The main result presented in the manuscript is the increase in capacitance between electrodes caused by the deposition of Ca on the trap surface. The capacitance increase is consistent with a theoretical prediction and preciously observed measurements of ions in surface traps [8]. The results are of general interest to the ion trapping community, and the technique has not been presented elsewhere in the context of anomalous heating. For these reasons, I feel that the work could be published, however, the current manuscript does not warrant publication.
The manuscript is in need of significant revisions and lacks sufficient data to support the conclusions. While the authors present a sufficient amount of data to support their claim that the adatom thickness leads to an increase in the capacitance, they do not make a concrete connection to anomalous heating. The manuscript lacks any ion-based measurements that would solidify the connection between the increase of adatoms on the electrode surface and an increase in motional heating. I feel that for the result to be useful to others in the field, ion-based measurements must be included. I will remind the authors that the influence of surface contamination on anomalous heating is still an open question in the ion trapping community and a measurement of surface contamination is not sufficient to draw conclusions about the expected anomalous heating rate.
In addition, the English in the manuscript needs to be improved significantly before it is published in this or any other journal.
For these reasons, I do not recommend publication of the manuscript in the current form and feel that additional measurements are required in order for the work to be published.
